# Energy Criticality Avoidance-Based Delay Minimization Ant Colony Algorithm for Task Assignment in Mobile-Server-Assisted Mobile Edge Computing

**DOI:** 10.3390/s23136041

**Published:** 2023-06-29

**Authors:** Xiaoyao Huang, Bo Lei, Guoliang Ji, Baoxian Zhang

**Affiliations:** 1Research Institute China Telecom, Beijing 102209, China; 2No. 208 Research Institute of China Ordnance Industries, Beijing 102227, China; 3University of Chinese Academy of Sciences, Beijing 100049, China; bxzhang@ucas.ac.cn

**Keywords:** mobile edge computing, mobile servers, task assignment, energy use balancing

## Abstract

Mobile edge computing has been an important computing paradigm for providing delay-sensitive and computation-intensive services to mobile users. In this paper, we study the problem of the joint optimization of task assignment and energy management in a mobile-server-assisted edge computing network, where mobile servers can provide assisted task offloading services on behalf of the fixed servers at the network edge. The design objective is to minimize the system delay. As far as we know, our paper presents the first work that improves the quality of service of the whole system from a long-term aspect by prolonging the operational time of assisted mobile servers. We formulate the system delay minimization problem as a mixed-integer programming (MIP) problem. Due to the NP-hardness of this problem, we propose a dynamic energy criticality avoidance-based delay minimization ant colony algorithm (EACO), which strives for a balance between delay minimization for offloaded tasks and operational time maximization for mobile servers. We present a detailed algorithm design and deduce its computational complexity. We conduct extensive simulations, and the results demonstrate the high performance of the proposed algorithm compared to the benchmark algorithms.

## 1. Introduction

With the rapid development of the 4G/5G cellular mobile communication networks, the computing demand for real-time applications of the Internet of Things and mobile users is dramatically increasing [1,2]. It is predicted that 41.8 billion intelligent terminal devices will be connected to the network in 2025 for intelligent monitoring, intelligent manufacturing, the Internet of Vehicles, etc. [3,4,5,6], and the level of global device data will reach 73.1 ZB [7,8]. Task processing via traditional centralized cloud computing architecture would lead to high delays and serious network congestion [9,10]. Therefore, traditional cloud computing architecture cannot meet the requirement of time-delay-sensitive applications, and mobile edge computing (MEC) has been proposed to tackle these issues, and has been regarded as a supplement of the cloud computing network [11,12,13].

However, the spatiotemporal dynamics of user requests make the fixed architecture of computing resource deployment at the network edge often unable to meet the quality of service (QoS) requirements of tasks [14,15]. Therefore, it is of great significance to introduce mobile computing resources to the traditional MEC system to achieve improved QoS [16,17]. However, mobile devices providing mobile computing services are usually battery-powered and have limited energy, and thus, will be reluctant to provide computing services when their residual energy falls below their personalized self-reserved energy threshold [18,19,20,21]. For this reason, it is of great significance to provide users with continuously satisfactory services by jointly optimizing the energy management of the mobile edge servers and the task allocation of the edge system to minimize the task delay and maximize the duration of continuous service of the mobile edge servers.

In this paper, we study the joint optimization of task allocation and energy management in a mobile-server-assisted MEC system. The design objective is to minimize the system delay while maximizing the operational time of the cooperative system. This is a critical issue for enabling effective mobile edge computing services. Considering the limited energy at mobile servers, this paper introduces the energy criticality threshold to dynamically determine the energy-critical mobile servers and prevents them from participating in task processing, so as to effectively extend the overall service time of mobile computing resources. Based on this concept, we design a dynamic energy-criticality-based delay minimization ant colony algorithm, considering the load, the bandwidth of each server, and the residual energy of the mobile servers to minimize the system delay while effectively prolonging the service time of the mobile servers by optimizing the task assignment and energy management. To the best of our knowledge, our paper is the first work addressing how to increase the system’s overall quality of service over the long term by prolonging the lifetime of assisted mobile servers. The main contributions of this paper are as follows:We build a mobile-server-assisted edge computing framework where tasks can be offloaded onto local fixed edge servers, remote fixed servers, or mobile servers.We formulate the task offloading problem for delay minimization as a mixed-integer programming (MIP) problem and prove its NP-hardness.We design a dynamic energy-criticality-based ant colony algorithm to address the above problem. We present a detailed algorithm design and deduce its computational complexity.We conduct extensive simulations, and the results show the high performance of the proposed algorithm as compared with benchmark algorithms.

The remainder of this paper is organized as follows: Section 2 briefly reviews related work. Section 3 describes the system model and formulates the problem under study. In Section 4, we design a dynamic energy-criticality-based ant colony algorithm to solve the formulated problem. In Section 5, we conduct extensive simulations for performance evaluation. In Section 6, we conclude this paper.

## 2. Related Work Literature Review

In this section, we will give a brief review of existing work in the area of task assignment in MEC. The existing work for task assignment in MEC can be divided into vertical and horizontal offloading according to the offloading direction. Different types of task offloading have different requirements according to the characteristics of the network architecture and resource distribution. In addition, delay and energy are two important optimization measures for task offloading that result in different strategies.

Vertical task offloading (e.g., [22,23,24,25,26,27]) typically considers a multi-level network architecture and optimizes the task assignment via cloud–edge collaboration. In [22,23,24,25,26,27], the central cloud is taken as a collaborative supplement to the edge, and tasks beyond the edge’s service capacity will be further offloaded onto the central cloud. As a result, the focus is to optimize the offloading decisions between the edge cloud and central cloud, while improving the service quality of the system. However, tasks offloaded onto the central cloud usually experience a longer delay than the edge, so this edge–cloud collaborating approach essentially expands the overall service capacity of the edge system by providing a degraded service quality.

Horizontal task offloading refers to the task offloading among servers at the edge layer. The use of mobile devices with idle resources for collaborative task processing at the edge layer has been considered to improve the service capability of the edge layer [28,29,30]. The authors of [28] studied the problem of collaborative offloading between multiple edge servers, each of which hosts different applications. They studied the joint load balancing and collaborative offloading between different edge servers without considering the use of mobile servers for improved service performance. The authors of [29,30] studied the multi-user task offloading problem in a D2D-assisted collaborative MEC network. They considered the joint optimization of channel resource allocation and offloading decisions for achieving maximal total utility for all users, which is to achieve a good balance between latency and energy consumption. In [29,30], the D2D-assisted offloading is restricted to one-hop D2D neighbors, which limits the service-enhancing capability. In [31,32,33], parked vehicles are considered as collaborative computing resource entities. In these papers, the edge service platform works to optimize the task scheduling and resource allocation based on the available resources at the edge as well as collaborative parked vehicles for improved resource utilization. However, none of the above studies considered the impact of limited energy at collaborative entities, and further, how it will affect the system performance.

Energy consumption is an important metric for mobile edge computing, which usually includes transmission energy consumption and computing energy consumption. The authors of [34,35,36,37] studied the task offloading in MEC with UAV assistance by optimizing both the task offloading and trajectory control to minimize energy consumption. In [38], the problem of minimizing terminal energy consumption under strict delay constraints was investigated. The authors of [39] studied how to minimize the total system energy consumption by optimizing task allocation under the edge computing framework while meeting delay constraints. In addition, delay is the most typical metric for MEC systems, as MEC applications are usually delay-sensitive. The authors of [40] studied the joint optimization of computing offloading and wireless transmission scheduling with multi-user resource competition to minimize the system latency. The authors of  [41] jointly considered the computation offloading, content caching, and resource allocation as an integrated model to minimize the total latency consumption of the computation tasks. In [42], the latency minimization problem was investigated by jointly coordinating the task assignment, computing, and transmission resources among the edge devices, multi-layer MEC servers, and the cloud center. The authors of [43] considered scenarios where the amount of data varies over time and application, and divide the system into two states based on data generation speed: non-blocking and blocking. Through joint optimization of task scheduling, computation, and communication resource allocation, delay is minimized in both states. However, none of the above papers consider the long-term minimization of system delay by prolonging the operational time of assistant mobile servers.

In summary, all the above studies fall into the category of collaboration-based MEC focused on minimizing the energy consumption of user devices (or the system as a whole) without considering the energy consumption of mobile servers, and further, how to prolong the working time of the battery-operated mobile servers in this case. Furthermore, due to the diversity of available energy, workload, and transmission conditions at different mobile servers, simply minimizing the energy consumption of individual mobile servers does not necessarily lead to global optimal energy use for a collaboration-based MEC network. In this paper, we study the joint optimization of task allocation and energy consumption management in a mobile-server-assisted MEC system, in order to minimize the task delay while maximizing the working time of the cooperative system.

## 3. System Model and Problem Formulation

In this section, we first describe the system model under study in this paper, including the network model, task model, and energy consumption model. Then, we formulate the problem under study in this paper. Table 1 lists the major notations used in this paper.

### 3.1. Network Model

Figure 1 shows the mobile-server-assisted MEC system under study in this paper, which consists of a managing platform; a set of fixed edge servers, denoted by N={1,2,…,N}; a set of mobile servers, denoted by M={1,2,…,M}; and a set of mobile users, denoted by K={1,2,…,K}. The managing platform works to maintain the system status, collect user requests, and accordingly make optimized task offloading decisions. Each fixed edge server is co-located with a base station (BS) and connected via a high-speed wired link. Thus, the transmission delay between them can be ignored. Thus, unless otherwise stated, we shall use symbol n∈N to represent both a base station, and also the server co-located with the base station, hereafter. Each base station corresponds to a cellular cell, where some users and mobile servers reside. In this paper, cells are assumed to be non-overlapping. Fixed edge servers can provide computing services for mobile users. In addition, fixed servers are connected to each other through metropolitan area network (MAN) optical fibers, which have high-speed transmission rates, and can be regarded as a single-hop completely connected network. Mobile servers are mobile devices with rich–idle computing resources (e.g., intelligent wireless terminals and intelligent vehicles), which are in the coverage of the base stations and reachable via cellular links, and can assist the fixed edge servers to provide computing services to the mobile users. The set of all mobile and fixed servers is denoted by S=N∪M={1,2,…,S}, where *S* is the total number of mobile and fixed servers. Furthermore, let $f_{s}$ represent the computing capacity of server s∈S, which is assumed to be heterogeneous.

The system works in a cycle-by-cycle fashion. A time cycle is represented by t∈T={1,…,T}. At the beginning of each cycle, it is assumed that the management platform knows the information of all tasks in the cycle. Let $R^{t}$ represent the set of tasks that arrive in time cycle *t*. Each user k∈K can generate multiple tasks in one cycle. To ease the exposition, a user with multiple tasks is treated as multiple (virtual) users, so that the user set and the task set are in one-to-one correspondence. Let $r_{k}\in R^{t}$ represent the task generated by user k∈K, then, we have the task set $R^{t}=\left\{r_{1},\ldots ,r_{k}\right\}$. Each task can be denoted by a triple $<C_{k},D_{k},t_{k}>$, where $C_{k}$ represents the task computation workload, $D_{k}$ represents the task data size, and $t_{k}$ is the task generation time. The mobility model of mobile servers and mobile users is based on the point of interest (POI) model, such that each of them has its own POI set. The model records each time a mobile user/server stays at one POI for a random period of time, and randomly moves to another POI. The set of collaborative mobile servers for a fixed server n∈N under the coverage of a same base station in the time cycle *t* is denoted as $M_{n}^{t}\subseteq M$.

Mobile servers are powered by batteries with different capacities, and each of them has to reserve a personalized amount of energy to meet its basic operating needs. As a result, when the residual energy is below the personalized threshold, a mobile server will stop providing the assisted computing services. Generally, the managing platform optimizes task assignment according to task information, status of edge servers, and network connection status, combined with the energy status of mobile servers, to maximally prolong the working lifetime of mobile servers by dynamically balancing energy consumption among them in order to minimize the system delay.

### 3.2. Delay Model

In the mobile-server-assisted edge network under study, each task $r_{k}\in R^{t}$ in time cycle *t* can be offloaded in one of the following four modes.
Directly offloaded onto the local fixed edge server n∈N for processing;Offloaded onto a mobile server $m\in M_{n}^{t}$ in the same cell via the relaying of its associated base station *n* via two hops of wireless cellular network link (Note that D2D connection is not considered in the scenario studied in this paper); Offloaded onto a remote fixed edge server $n^{\prime }\in N\setminus \left\{n\right\}$ via the relaying of its associated base station *n* via one-hop wireless link of accessing *n* and one-hop wired link to the remote fixed edge server;Offloaded onto a remote mobile server $m\in M_{n^{\prime }}^{t}$ via path “local base station–remote base station–remote mobile server” via the concatenation of wireless link and wired link.

For the above four offloading modes, the transmission delay experienced by a task has the following cases, respectively: For the first case, it equals the one-hop wireless link transmission delay from the user to their associated base station (i.e., the local fixed server). For the second case, it equals a two-hop wireless path transmission delay from the user to a local mobile server via their associated base station. For the third case, it equals the sum of the wireless link transmission delay from the user to their associated base station, and the wired link transmission delay from their associated base station to a remote base station where the remote fixed server resides. For the fourth case, it equals the sum of transmission delay of two wireless links and one wired link. Because all the fixed servers are connected via high-speed wired links, they form a fully connected network. Thus, the length of the longest offloading path is three hops. In this paper, we assume that task results are very small, and thus, their delivery delays are considered negligible.

Let $a_{k,s}$ denote whether task $r_{k}\in R^{t}$ is assigned to a task executor s∈S. We have
(1)$$a_{k,s}=\left\{\begin{array}{cc} 1 & \mathrm{if}\;\mathrm{task}\;r_{k}\in R^{t}\;\mathrm{is}\;\mathrm{assigned}\;\mathrm{to}\;\mathrm{server}\;s\in S\\ & \mathrm{for}\;\mathrm{processing}\\ 0 & \mathrm{otherwise}. \end{array}\right.$$

The delay of a task $r_{k}\in R^{t}$ for offloading to a server s∈S, denoted by $T_{k,s}^{total}$, typically includes transmission delay, waiting delay, and processing delay, which can be calculated as follows:(2)$$T_{k,s}^{total}=T_{k,s}^{trans}+T_{k,s}^{com}+T_{k,s}^{wait}.$$

The processing delay $T_{k,s}^{com}$ can be calculated as follows:(3)$$T_{k,s}^{com}=\frac{C_{k}}{f_{s}}.$$

In this paper, we assume that tasks are processed at the server side in a first-in-first-out fashion. Thus, the waiting delay of a task $r_{k}$ at a server *s* is jointly decided by the set of unprocessed tasks $R_{s,left}^{t}$ at the beginning of time cycle *t*, which were assigned in the preceding time cycles but have not been processed yet, and the set of tasks $R_{s,ahead}^{t}$, which were assigned in this time cycle but ahead of task $r_{k}$. As a result, $T_{k,s}^{wait}$ can be calculated as
(4)$$T_{k,s}^{wait}=\frac{R_{s,left}^{t}+R_{s,ahead}^{t}}{f_{s}}.$$

The transmission delay of task $r_{k}$ (denoted by $T_{k}^{trans}$) includes the transmission delay of the wireless link(s) and the transmission delay of wired link used (if any). Time division multiplexing and first-in-first-out transmission scheduling are assumed to be adopted for task transmissions, and there is only one task in transmission over a link at any time. Let the binary variables $h_{k,n}^{t}$ and $h_{m,n}^{t}$ represent whether the user *k* and the mobile server *m* are under the coverage of BS *n*, respectively. Moreover, let binary variables $x_{s,wl}^{\delta }\left(r_{k}\right)$ and $x_{s,lt}^{\delta }\left(r_{k}\right)$ represent whether it is suitable to start transmitting task $r_{k}\in R^{t}$ in time slot δ∈t via a wireless link and a wired link associated with server *s*, respectively. For all the above binary variables, if their corresponding conditions hold, their values are 1 or otherwise 0. Let decision variables $y_{k,k^{\prime }}^{wl}\left(s\right)$, and $y_{k,k^{\prime }}^{lt}\left(s\right)$ represent the transmission sequence of two tasks $r_{k}$ and $r_{k^{\prime }}$ through wireless and wired links associated with the server *s*, respectively: If task $r_{k}$ is transmitted before task $r_{k^{\prime }}$, its value is 1, otherwise it is 0.

Thus, the transmission delay for each of the above four offloading modes can be accordingly calculated. For the first offloading mode, the transmission delay of task $r_{k}$ is the transmission delay of the wireless link from user *k* to its associated BS *n*, which is calculated as
(5)$$T_{k,n}^{trans}=\left(\sum_{r_{k^{\prime }}\in R^{t}\cup r_{k}}y_{k^{\prime },k}^{wl}\left(n\right)\frac{D_{k’}}{Dr_{k’,n}}\right),$$
where the right side of the above equation represents the total transmission time needed for those tasks to arrive before task $r_{k}$ plus $r_{k}$. $Dr_{k,n}$ is the data rate from user *k* to server *n*, and it is calculated as
(6)$$Dr_{k,n}=w_{n}\log_{2}\left(1+\frac{P_{k}H_{k,n}}{\sigma^{2}}\right),$$
where $w_{n}$ is the bandwidth of BS *n*, $P_{k}$ is the transmission power of user *k*, $H_{k,n}$ is the wireless channel gain between user *k* and BS *n*, and $\sigma^{2}$ is ambient noise.

For the second offloading mode, following that for the first offloading mode, the total transmission delay will increase by one-hop wireless transmission delay from BS *n* to the chosen local mobile server *m*, which is calculated as $T_{n,m}^{trans}=\frac{D_{k}}{Dr_{n,m}},m\in M_{n}^{t}$, where the data rate from BS *n* to the mobile server *m* is $Dr_{n,m}=h_{m,n}^{t}w_{n}\log_{2}\left(1+\frac{P_{n}H_{n,m}}{\sigma^{2}}\right)$.

For the third offloading mode, following that for the first offloading mode, the total transmission delay will increase by one-hop wired transmission delay from BS *n* to BS $n^{\prime }$, which is calculated as
(7)$$T_{n,n^{\prime }}^{trans}=\left(\sum_{r^{k^{\prime }}\in R^{t}\cup r^{k}}y_{k^{\prime },k}^{lt}\left(s\right)\frac{D_{k}}{v_{n,n^{\prime }}}\right),$$
where the first item in the right side of the above equation represents the waiting time for task $r_{k}$, the second item represents the transmission time of the task $r_{k}$ on the wired link, and $v_{n,n^{\prime }}$ is the transmission data rate of the link. Here, we assume that the data rate of wired links connecting fixed servers is much higher than that of a wireless link between a base station and users, so there is no queueing issue for the transmission over a wired link.

For the forth offloading mode, following that for the third offloading mode, there will be an additional transmission delay for the transmission from BS $n^{\prime }$ to the chosen mobile server $m^{\prime }$, denoted as $Trans_{n^{\prime },m^{\prime }}^{wl}$, whose calculation is similar to that for the downlink transmission in the second offloading mode.

In summary, the delay of the task $r_{k}$ to its assigned server *s* can be represented as $TD_{k}=\sum_{s\in S}a_{k,s}T\;D_{k,s}$.

### 3.3. Energy Consumption Model

Mobile servers are powered by batteries with different capacity constraints, and have to reserve personalized levels of energy to maintain self-working. A mobile user will quit the assisted computing service when its residual energy is less than the required reserved amount of energy. Fixed servers are powered by a power grid with a stable energy source, and thus can provide a continuous computing service. As a result, considering the energy balancing among mobile servers during the process of task assignment to prolong the serving time of mobile servers is significant to reduce system service delay by ensuring the overall service capability of the whole MEC system for as long as possible. For the mobile server *m*, we use $E_{m}^{max}$ and $E_{m}^{sel\;f}$ to denote the initial amount of energy and the amount of personalized reserved energy, respectively. The energy consumption of the processing task $r_{k}$ by the mobile server *m* can be calculated as
(8)$$E_{m,k}^{cm}=a_{k,m}\gamma C_{k}f_{m}^{2},$$
where γ is the energy consumption coefficient determined by the structure of the chip. At the beginning of the time cycle *t*, the residual energy of the mobile server *m* denoted by $E_{m}^{t}$ can be calculated as
(9)$$E_{m}^{t}=E_{m}^{max}-\sum_{t^{\prime }=1}^{t-1}\sum_{r_{k}\in R^{t^{\prime }}}a_{k,m}E_{m,k}^{cm},$$
where the left part is the initial power, and the right part is the amount of energy consumption of the tasks assigned to *m* before the time cycle *t*.

### 3.4. Problem Formulation

Delay is one of the most important performance metrics for an MEC system. Moreover, in the mobile-server-assisted MEC system under study in this paper, it is also important to optimize the energy use balancing among mobile servers, which can effectively extend the service time of the mobile servers, and thus ensure high service capacity of the collaborative MEC network. As a result, considering the heterogeneity of different fixed servers and mobile servers, and also the spatiotemporal task distribution, this paper aims to minimize the system delay by optimizing task assignment and also energy use balancing under the energy constraint of mobile servers.
(10a)$$\begin{array}{cc} Minimize & \sum_{t\in T}\sum_{r_{k}\in R^{t}}TD_{k}\;\; \end{array}$$
(10b)$$\begin{array}{cc} s.t. & E_{m}^{t}\geq E_{m}^{sel\;f},\forall t\in T,\forall m\in M; \end{array}$$
(10c)$$\begin{array}{cc} & \sum_{s\in S}a_{k,s}=1,\forall r_{k}\in R^{t},t\in T; \end{array}$$
(10d)$$\begin{array}{cc} & \sum_{r_{k}\in R^{t}}x_{s,wl}^{\delta }\left(r_{k}\right)\leq 1,\sum_{r_{k}\in R^{t}}x_{s,lt}^{\delta }\left(r_{k}\right)\leq 1,\forall s\in S,\delta \in t; \end{array}$$
(10e)$$\begin{array}{cc} & \sum_{\delta \in t}x_{s,wl}^{\delta }\left(r_{k}\right)\leq 1,\sum_{\delta \in t}x_{s,lt}^{\delta }\left(r_{k}\right)\leq 1,\forall s\in S,r_{k}\in R^{t};\\ & \sum_{\delta \in t}x_{s,wl}^{\delta }\left(r^{k^{\prime }}\right)\delta \leq \sum_{\delta \in t}x_{s,wl}^{\delta }\left(r^{k}\right)\delta +h_{k,n}^{t}\frac{D_{k}}{Dr_{k,n}} \end{array}$$
(10f)$$\begin{array}{cc} & \;\;\;\;-H(1-y_{k,k^{\prime }}^{wl}\left(s\right));\\ & \sum_{n\in N}h_{k,n}^{t}=1,\forall k\in K,t\in T,\sum_{n\in N}h_{m,n}^{t}=1, \end{array}$$
(10g)$$\begin{array}{cc} & \;\;\;\;\forall m\in M,t\in T. \end{array}$$
In this instance, constraint (10b) is the energy constraint for mobile servers, such that the residual energy of a mobile server at any time should be higher than or equal to its personalized reserved amount. Constraint (10c) ensures that each task is assigned to only one server. Constraint (10d) represents that the number of tasks that start to be transmitted over a wired/wireless link should be no more than one at a time. Constraint (10e) ensures that a task can only be transmitted once on the server associated link. Constraint (10f) guarantees that a task can only start to be transmitted when the transmission of the preceding task is finished when they share the same link. Constraint (10g) represents that each user or mobile server is associated with only one BS during a time cycle.

The formulated task assignment problem can be reduced to the classical Job-Shop problem, and thus, NP-hard. To address this issue, in this paper, we propose an efficient heuristic algorithm—dynamic energy-criticality-based delay minimization ant colony algorithm (EACO)—by optimized task assignment and energy use balancing.

## 4. Proposed EACO Algorithm

In this section, we propose the EACO algorithm, which works to optimize the task assignment while pursuing the minimal system delay in a cycle-by-cycle fashion. EACO models the assignment of tasks arrived in a cycle to the set of servers as a directed multistage graph for finding optimized “task”server‘’ matches. EACO searches the directed graph by using multiple ants in multiple rounds. Different ants independently traverse the graph in a probabilistic way based on the pheromone information left at different nodes, i.e., the nodes’ residual energy statuses, as well as their personalized reserved energy requirements. To maximally utilize the service capacity of mobile servers, EACO works to balance the energy consumption among mobile servers subject to their personalized reserved energy constraints. Each path taken by an ant corresponds to a feasible solution, and servers on the path taken by an ant will be left with a certain amount of pheromone for guiding the ant-based search in the next round. The above process repeats until an expected number of rounds is reached, and among all the choices, the assignment leading to the minimum system delay is selected.

In the following, we will show (1) how to construct the directed multistage graph for each cycle, and also how to calculate the transition probability at each node in the graph; (2) how to initialize and update the pheromone at nodes; (3) how to dynamically protect those energy-critical mobile servers for energy use balancing, and also, maximizing their operational time; and finally, we present the detailed design of EACO.

### 4.1. Directed Multistage Graph Construction and Transition Probability Calculation

EACO works in a cycle-by-cycle fashion for task assignment. In EACO, the assignment of tasks $R^{t}=\left\{r_{1},\ldots ,r_{K}\right\}$ that arrive in time cycle *t* to the server set S={1,…,S} is modeled as a directed multistage graph (see Figure 2). There are *K* layers in the graph, plus an extra “Start” node, and each layer corresponds to a task, and contains *S* servers as candidate task processors for the task. All ants start from the “Start” node, and each of them independently selects a node in the next layer in a probabilistic way as the processor for the corresponding task. This process continues until the last layer. Accordingly, each ant brings a feasible task assignment solution.

For the directed multistage graph built for each cycle, EACO searches it for *G* rounds in total, and issues $A^{Num}$ ants in each round. Let <k,s> represent the blue vertex located in the kth layer and sth column in the graph, 1≤k≤K,1≤s≤S. If an ant passes through the vertex <k,s>, task $r_{k}$ is assigned to the server *s*. Once the task $r_{k-1}$ is assigned (Here, we treat the “Start” node as task $r_{0}$, which is null.), the assignment for task $r_{k}$ is to be determined. In EACO, the next vertex for an ant is determined by using the roulette method based on the transition probability $P_{k,s}^{b}\left(g\right)$ of each candidate vertex <k,s> (see Equation (Equation 11)). The detailed procedure for roulette-based vertex selection is as follows: First, the transition probability $P_{k,s}^{b}\left(g\right)$ of each vertex is mapped to the hit probability on the roulette, i.e., ${\hat{P}}_{k,s}^{b}\left(g\right)=\frac{P_{k,s}^{b}\left(g\right)}{\sum_{s\in S}P_{k,s}^{b}\left(g\right)}$. Then, a random value *p* in interval (0,1) is generated. After that, each time a random node is chosen from the remaining candidate vertex set until reaching a vertex <k,j>, such that the accumulated probability is $\hat{P}\left(j\right)=\sum_{s=1}^{j}{\hat{P}}_{k,s}^{b}\left(g\right)\geq p$ for the first time, in which case, the vertex <k,j> is chosen as the vertex in the next layer.

The transition probability $P_{k,s}^{b}\left(g\right)$ of ant *b* choosing the vertex <k,s> in the gth round is calculated as follows:(11)$$P_{k,s}^{b}\left(g\right)=\left\{\begin{array}{cc} \frac{{\left[\tau_{k,s}\left(g\right)\right]}^{\alpha }{\left[\eta_{k,s}\left(g\right)\right]}^{\beta }}{\sum_{s’\in B^{allowed}}{\left[\tau_{k,s’}\left(g\right)\right]}^{\alpha }{\left[\eta_{k,s’}\left(g\right)\right]}^{\beta }} & s\in B^{allowed}\\ 0 & \mathrm{otherwise}, \end{array}\right.$$
where $B^{allowed}$ is the set of nodes (servers) still available for the ant *b* to assign tasks. For a mobile server, if its current remaining energy is less than its personalized reserved energy level, it will be removed from the set $B^{allowed}$, and thus out of consideration. $\tau_{k,s}\left(g\right)$ is the pheromone information of vertex <k,s>, and α is the pheromone factor. $\eta_{k,s}\left(g\right)$ is an energy criticality avoidance function for exempting those energy-critical mobile servers from providing offloading services, thus achieving prolonged operational time. β is a constant factor. In EACO, pheromone updating and energy criticality avoidance are used together to guide the ant-based search for optimized task assignment.

### 4.2. Pheromone Initialization and Updating

When an ant walks to the end of a path in the multistage graph, which means that all the task assignments are determined, the pheromone will be left at each vertex on the path, and the pheromone value at each vertex on the path needs to be updated accordingly. Let $\Delta \tau_{k,s}\left(g\right)$ denote the pheromone increment of vertex <k,s>. Whether ant *b* passes through vertex <k,s>, $\Delta \tau_{k,s}\left(g\right)$ is calculated as follows:(12)$$\Delta \tau_{k,s}^{b}\left(g\right)=\left\{\begin{array}{cc} \frac{Q}{TD_{sys}^{b}\left(g\right)} & \mathrm{if}\;\mathrm{ant}\;b\;\mathrm{passes}\;\mathrm{through}\\ & \mathrm{vertex}\;<k,s>;\\ 0 & \mathrm{otherwise}, \end{array}\right.$$
where $TD_{sys}^{b}\left(g\right)$ is the system delay as achieved by ant *b* in this round, and *Q* is a constant used for characterizing the total amount of pheromone.

After the gth (1≥g≥G) round is complete, according to the new pheromones left by all the ants in this round, and also the evaporation of existing pheromones, global pheromone updating is needed for preparation of the next (g+1)th round. Specifically, the global pheromone at each vertex <k,s> is updated as follows:(13)$$\tau_{k,s}\left(g+1\right)=\left(1-\rho \right)\tau_{k,s}\left(g\right)+\Delta \tau_{k,s}\left(g\right),$$
where $\Delta \tau_{k,s}\left(g\right)=\sum_{b=1}^{A^{Num}}\Delta \tau_{k,s}^{b}\left(g\right)$ is the total amount of pheromone left at vertex <k,s> by all the ants in round *g*, $A^{Num}$ is the total number of ants, and ρ is the pheromone concentration volatilization coefficient.

In addition, due to server heterogeneity, different servers may have different processing capabilities and bandwidth resources. Therefore, the setting of the initial pheromones for different servers should consider their heterogeneous resource richness, which is helpful for accelerating the optimization speed. Consequently, the pheromones are deferentially initialized according to the computing capacities and the available data rates at different servers. Following this, the data rate available to each mobile server is calculated. The pheromone initialization for a server *s* is as follows:(14)$$\tau_{k,s}\left(0\right)=h_{1}f_{s}+h_{2}B_{s},$$
where $h_{1}$ and $h_{2}$ are the coefficients for computing resources and available data rates of servers, respectively.

### 4.3. Dynamic Energy Criticality Avoidance

In EACO, when a mobile server’s residual energy drops below its personalized reserved energy level, it will stop providing offloading services. Thus, the operational time of mobile servers affects their service continuity. To maximally prolong the operational time of the mobile servers as a whole, we introduce the concept of energy criticality. That is, a mobile server whose residual usable energy is in the energy-critical range (e.g., those mobile servers with the lowest ξ percent residual usable energy, 0≥ξ≥100, typically, ξ is a small number, e.g., 10–15) is said to be energy-critical. More specifically, mobile servers are first sorted in ascending order of their residual usable energy $E_{m,o}=E_{m}^{lef\;t}-E_{m}^{sel\;f}$. Accordingly, those mobile servers whose residual usable energy is in the lowest ξ% will be added to the energy-critical mobile server set, denoted by $M^{protected}$ (The performances of two metrics for sorting, including residual energy and the ratio of residual energy and total providable energy, are compared by simulations. The results show that sorting according to the value of residual energy has better effects in terms of minimizing delay and quitting rate of mobile servers. Therefore, this paper adopts the method of sorting based on residual energy). In EACO, a mobile server in energy-critical status is exempted from providing offloading service. It should be noted that as those in-service mobile servers keep burning their energy for providing offloading services, the set of energy-critical mobile servers will change with time. Accordingly, we can maximally balance the energy use at different mobile servers in a dynamic manner, so as to maximize their operational time.

In the transition probability calculation in Equation (Equation 11), we introduce the energy criticality avoidance function, which works to avoid the involvement of energy-critical mobile server(s) for providing offloading services. Accordingly, the energy criticality avoidance function $\eta_{k,s}\left(g\right)$ for a task $r_{k}$ to be offloaded onto server *s* is calculated as follows:(15)$$\eta_{k,s}\left(g\right)=\left\{\begin{array}{cc} 0\; & s\in M^{protected}\\ \frac{1}{T\;D_{k,s}} & \mathrm{otherwise}. \end{array}\right.$$

In Equation (Equation 15), the transition probability of an energy-critical mobile server is set to zero. In contrast, for a non-critical energy mobile server *s* to undertake a task $r_{k}$, its transition probability is set as the reciprocal of the delay that the task experienced for its offloading to the server. As introduced earlier, the delay experienced by a task includes the transmission delay, waiting delay, and processing delay. According to the subpath that an ant has taken so far, and the corresponding task assignment associated with the subpath, the transmission delay for a subsequent candidate task can be calculated based on the already-assigned tasks in the transmission queue in the same cell. The waiting delay equals the total processing time of those tasks that have already been assigned to the server where the task is offloaded. The processing delay corresponds to the computing delay of the task at the server. Therefore, the transition probability for each next candidate vertex can be determined based on the task assignment associated with the subpath that an ant has taken so far.

### 4.4. Ant-Colony-Based Task Assignment

Based on the probability transfer function constructed earlier, the ant-colony-based task assignment algorithm can achieve near-optimal task assignment solution through multiple iterations of searching by multiple ants. Considering that fixed edge servers in general have high service capacity and unlimited energy, the first server that an ant chooses is always one of the fixed servers to reduce the randomness of ant optimization. The detailed procedure of the algorithm is presented in Algorithm 1. In Steps 1–3, the algorithm first initializes the ant colony, where minTime is the solution with the minimum delay so far, $ATime_{b}$ is the delay of the solution brought by ant *b*, Tabu is the set of solutions, and BestTabu is the best solution so far. For each candidate vertex, its initial pheromone is set according to Equation (Equation 14). After the initialization, the algorithm works iteratively to find an improved solution. First, each ant randomly chooses a fixed server as its starting vertex (Note that this choice only affects one of $\left|R^{t}\right|$ tasks. In general, the value of $\left|R^{t}\right|$ is large, so this setting has little impact on the overall solution.). Then, in each path selection process for task assignment, EACO determines the set of energy-critical mobile servers $M^{protected}$ (see Step 7). Accordingly, the energy criticality avoidance function $\eta_{k,s}\left(g\right)$ is updated and calculated by using Equation (Equation 15), so as to prevent those energy-critical nodes from participating in task assignment (see Step 8). According to the energy criticality avoidance function and the pheromone information, EACO calculates the probability $P_{k,s}^{b}\left(g\right)$ by Equation (Equation 11) of ant *b* choosing each candidate vertex in $B^{allowed}$, and then chooses one among them by using roulette (see Steps 9–10). After the assignment, the chosen vertex will be added to the partial solution $Tabu_{b}$, and the ant moves to the next layer (see Step 11).
**Algorithm 1:** EACO for task assignment. 
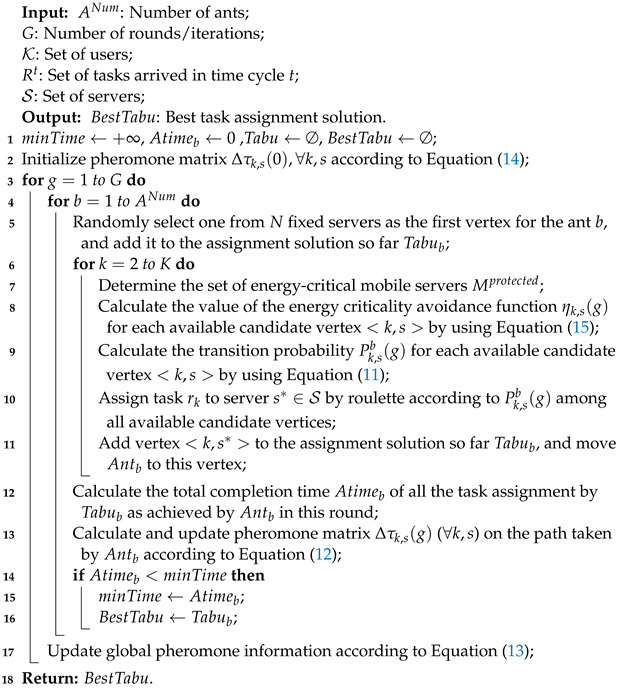


After all the ants complete their travels, the corresponding pheromone information $\Delta \tau_{k,s}\left(g\right)$ is updated according to Equation (Equation 12) (see Step 12). If an ant brings a better assignment solution, EACO updates minTime, and the best solution so far BestTabu (see Steps 13–16). When all ants have traversed the graph in this round, the global pheromone information is updated for the next round (see Step 17).

The complexity of Algorithm 1 can be deduced as follows: The complexity of the initialization in Step 2 is O(KS), while the complexity of the iteration process in Steps 3–17 is $O(GKSA^{Num})$, where the triple loops take $O(GKA^{Num})$ time and the server selection by roulette in Step 10 takes O(S) time. As a result, the overall complexity of Algorithm 1 is $O(GKSA^{Num})$.

## 5. Performance Evaluation

In this section, we conduct simulations to evaluate the performance of the proposed algorithm EACO. We adopt the open-source edge system emulator EdgeCloudSim [44] for this purpose, and develop a mobile-server-assisted edge network in the emulator for performance evaluation purposes.

### 5.1. Simulation Settings

We conduct simulations in a mobile-server-assisted edge computing system with 10 edge servers, 50 mobile servers, and 50∼300 users. The tasks generated by each user follow the Poisson distribution, and the rate is randomly chosen from [1,5] per second. The workload and data size of each task are random, within the ranges 1000∼10,000 (MI) and 1∼5 MB, respectively. The computation capacity of each fixed server is in the range 10,000∼30,000 (MIPS), and of each mobile server is in the range 5000∼8000 (MIPS). The initial amount of energy of each mobile server is random, within the range 20,000∼30,000 (kJ). Edge servers are connected through wired optical fiber, with a total bandwidth rate of 1300 Mbps. The mobile servers are connected to the BS through the cellular network, of which the bandwidth is 40 MHz and noise intensity is −50 dBm.The POI-based mobility model is adopted, and we set 1000 points of interest. Each point of interest is a randomly selected location within the coverage of a BS. Each user randomly stays at each location for a period of time. The number of ants and iterations for EACO are 50 and 100, respectively. The values of α, β, and ρ are 1.5, and 0.3, respectively. The total time cycles in a simulation are T=10 min, and each time cycle lasts 10 s.

### 5.2. Algorithms for Comparison

In the simulations, we consider different network architectures, and also different algorithms for comparison purposes.

In the simulations, the following two network architectures are considered:Mobile-server-assisted MEC network: The architecture under study in this paper;MEC network without assistance: In this architecture, only fixed edge servers are used to provide offloading services, without any assistance from mobile servers.

The benchmark algorithms for comparison are as follows:Random assignment (RandM): Each task is assigned randomly to a server s∈S for processing;Minimum workload first assignment (GreedyW): Each task is assigned to the server s∈S, whose workload is the minimum among all choices;Minimum delay first assignment (GreedyW): Each task is assigned to the server s∈S, whose resulting delay is the minimum among all choices.

### 5.3. Simulation Results

#### 5.3.1. The Impact of Number of Ants

The average system delay versus different ants is shown in Figure 3. In this experiment, the number of users is 300, the simulation duration is 10 min, the energy protection threshold is 0.2, and the number of iterations is 100. It can be seen that the system delay first decreases with the increase in the number of ants, and gradually slows down when the number of ants is 50. This is because more ants results in a higher degree of exploration in EACO to obtain a lower system delay, which will stop decreasing when EACO gradually converges to the optimal solution.

#### 5.3.2. The Impact of Energy Protection Threshold ξ

The impact of energy protection threshold ξ on system performance is shown in Figure 4. In this test, the number of users is 200, the duration of the simulation is 10 min, and the energy protection threshold varies within the range 0.05∼0.4. As shown in Figure 4a, it can be seen that the curve of the system delay reaches the lowest point when the energy protection threshold ξ equals 0.2, and both too-high or too-low energy protection thresholds result in a relatively large delay. The curve in Figure 4b presents a similar tendency as in Figure 4a. Figure 4b shows the curve of the quit rate of mobile servers of the system with varying ξ, and the quit rate is defined as the ratio of the number of exiting mobile servers due to the remaining energy being less than or equal to the required reserved energy of the mobile server itself. It can be seen that both too-high or too-low energy protection thresholds cause a relatively large system delay, and also a high quit rate of mobile servers; this is because protecting both too many or too few mobile servers cannot balance energy consumption among mobile servers effectively, and thus results in low resource utilization efficiency and relatively high system delay, accordingly.

#### 5.3.3. Comparison of Performance of Two Architectures

Next, we evaluate the performance of the proposed algorithm EACO when it works with and without the assistance of the ant colony algorithm for task assignment, where the latter is referred to as ACON. In this test, the simulation duration is 10 min, and the energy protection threshold is 0.2. As shown in Figure 5, the average delay of the system without assistance increases sharply as the number of users increases. In addition, the architecture based on mobile server assistance has obvious advantages when the number of users is large, and the performances of the two architectures are close when the number of users is small, for the reason that the existing fixed servers are enough for the system task processing of users.

#### 5.3.4. Comparison of Performance in Terms of Energy Protection

The average system delay and quit rate of mobile servers of EACO with different energy protection thresholds, which are set as 0.2 (EACO-0.2) and 0 (EACO-0), respectively, are compared as the simulation time increases. As shown in Figure 6a, the algorithm without energy protection EACO-0 achieves lower delay. With the increase in simulation time, the algorithm EACO-0.2 with energy protection has better performance than the algorithm EACO-0, and the gap becomes larger with the increase in time cycles. In addition, in Figure 6b, it can be seen that the quit rates of both algorithms increase with increasing simulation time, and the quit rate of EACO-0.2 is lower than that of EACO-0. At the initial phase of the system, the quit rate of mobile servers of EACO-0.2 is 0 and the delay is relatively large because EACO-0.2 protects energy-critical mobile servers, and only (1−ξ)% of mobile servers can be used for task processing. However, as simulation time increases, the role of energy consumption balancing in EACO-0.2 is reflected, which results in low quit rates of mobile servers, and thus, maintains a longer and higher service capacity with low overall system delay.

#### 5.3.5. Impact of Number of Users

The average delay and quit rate of mobile servers of different algorithms are simulated with varying number of users in Figure 7. In this test, the simulation time is 10 min and the energy protection threshold of EACO is 0.2. As shown in Figure 7a, the proposed algorithm EACO achieves the lowest delay, followed by GreedyT, GreedyW, and RandM. As the number of users increases, the delay of all the four algorithms increases correspondingly, but the increasing rate of EACO is lower than that of the other three comparison algorithms. The changing trends of the curves of GreedyW and RandM are close, and the gap between them is small, because random allocation also balances the load when the number of tasks is large. It can be seen from Figure 7b that the proposed algorithm EACO achieves the lowest quit rate of mobile servers, followed by the algorithms GreedyT, GreedyW, and RandM. As the number of users increases, the quit rate of all the four algorithms increases, and when the number of users is about 150, the quit rates of GreedyT, GreedyW, and RandM reach 1, meaning all mobile servers quit the collaboration after simulation.

#### 5.3.6. Comparison of Performance of Different Algorithms

The performance of the proposed algorithm EACO is compared with the other three benchmark algorithms with varying simulation time durations, as shown in Figure 8. In this test, the number of users is 300, and the energy protection threshold of EACO is 0.2. As shown in Figure 8a, the proposed algorithm EACO achieves the lowest delay, followed by GreedyT, GreedyW, and RandM. Furthermore, as simulation time increases, the gap between EACO and the three benchmark algorithms becomes larger, which indicates that the energy criticality avoidance strategy can improve the quality of service of the system in the long term. As shown in Figure 8b, the quit rate of the mobile servers produced by EACO is the lowest among all the simulated algorithms.

To summarize, we compare the delay performance of different algorithms and architectures in Table 2. Recall that EACO-0 means the implementation of EACO when the energy protection ratio ξ is set to 0. In Table 2, $A^{Num}=50$ and ξ=0.2 indicate that the optimal number of ants and the optimal energy protection ratio are 50 and 0.2, respectively, under our simulation setting. We take the delay achieved by the proposed algorithm EACO as the basic value 1, thus, the values of the (worst-case) delays by other algorithms and architectures can be represented by their ratios for visual comparison. For example, the delay caused by ACON is, in the worst case, 675% more than that by EACO, as observed at the end of the simulation time (see also Figure 8a). The results in Table 2 clearly show that EACO can greatly improve the delay performance as compared with benchmark algorithms and architectures.

## 6. Conclusions

In this paper, we studied the task assignment and energy management in mobile-server-assisted edge computing networks. By considering the server heterogeneity and energy constraints of mobile servers, we aimed at minimizing the system delay while prolonging serving time of mobile servers via energy use balancing. We formulated the system delay minimization problem as a mixed-integer programming problem. Due to the NP-hardness of this problem, we propose a dynamic energy criticality avoidance-based delay minimization ant colony algorithm. We define the transition probability for guiding the ant traversal by considering the energy criticality of mobile servers during the task assignment. We present an algorithm design and deduce its computational complexity. Extensive simulations were conducted, and the results demonstrate the high performance of the proposed algorithm.

The proposed algorithm in this paper is innovative, and introduces dynamic energy criticality avoidance to the classic ant colony algorithm. The deduced computational complexity and simulation results show that it achieves good performance with acceptable complexity. This paper focuses on the scenario that mobile servers work as the assistant processors for the fixed edge severs and cloud servers. For future directions, device-to-device offloading can also be considered for such assistant mobile servers; however, safety and privacy should be taken into consideration in such cases. In addition, the design of an effective incentive mechanism is also desirable for the mobile-server-assisted MEC services, which can be studied in future work.

## Figures and Tables

**Figure 1 sensors-23-06041-f001:**
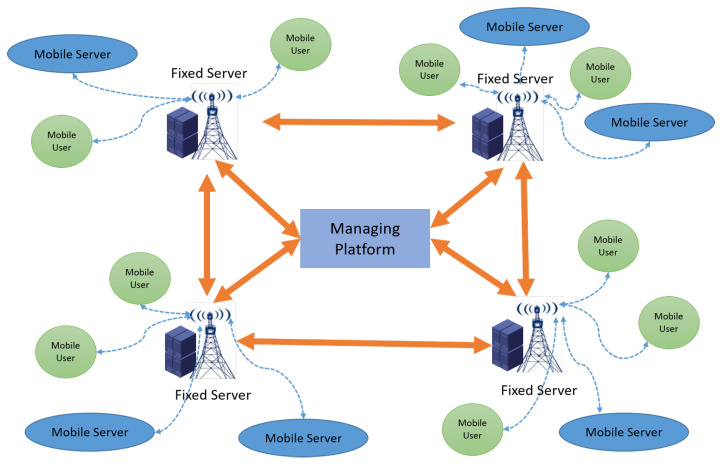
Architecture of the mobile edge computing system under study.

**Figure 2 sensors-23-06041-f002:**
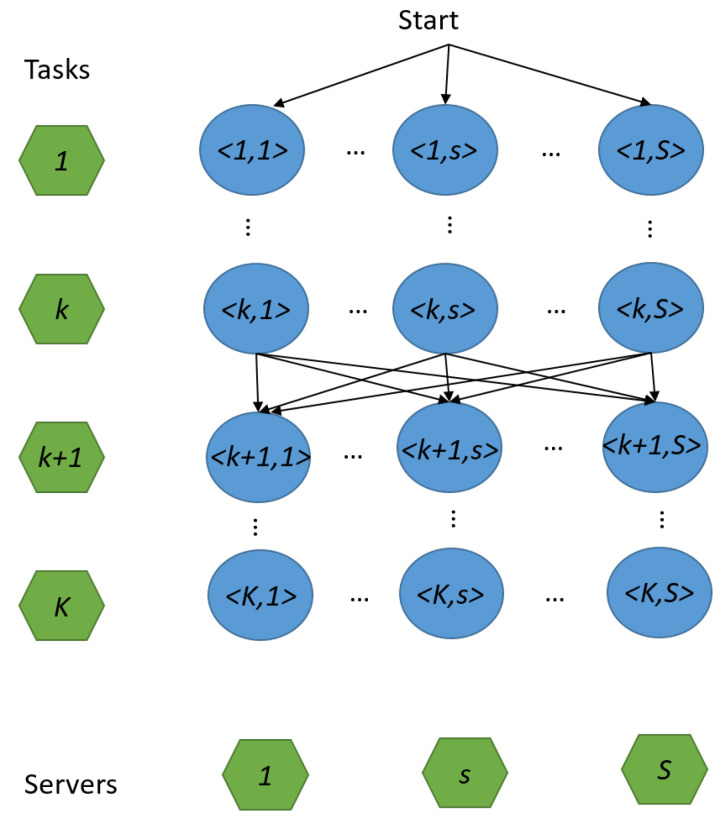
Directed multistage graph constructed for ant-based task assignment.

**Figure 3 sensors-23-06041-f003:**
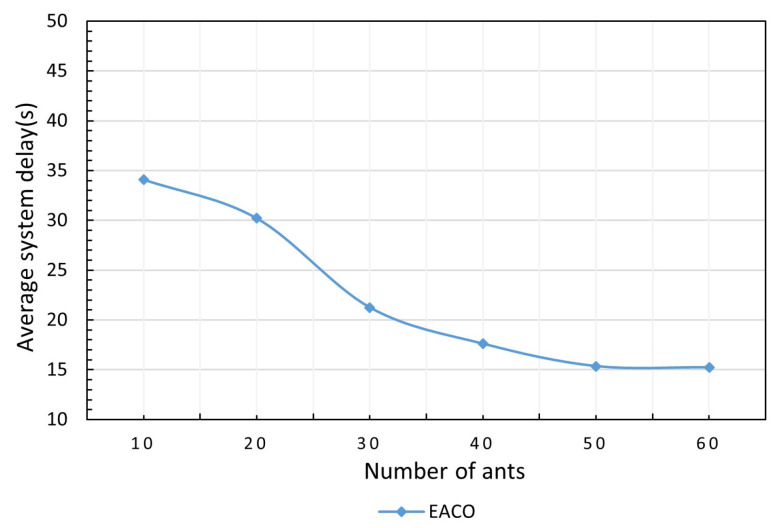
Impact of the number of ants on performance.

**Figure 4 sensors-23-06041-f004:**
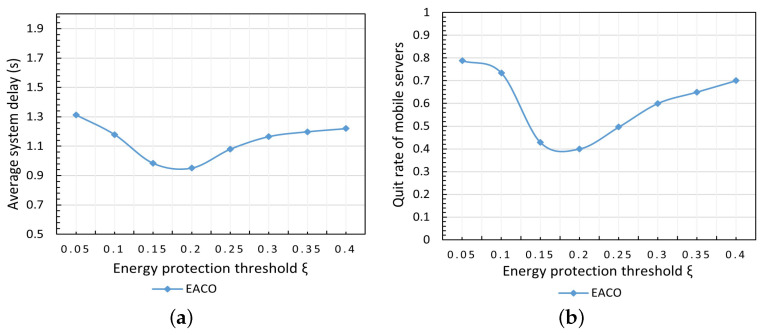
Impact of energy protection threshold ξ. (**a**) Average syst em delay. (**b**) Quit rate of mobile servers.

**Figure 5 sensors-23-06041-f005:**
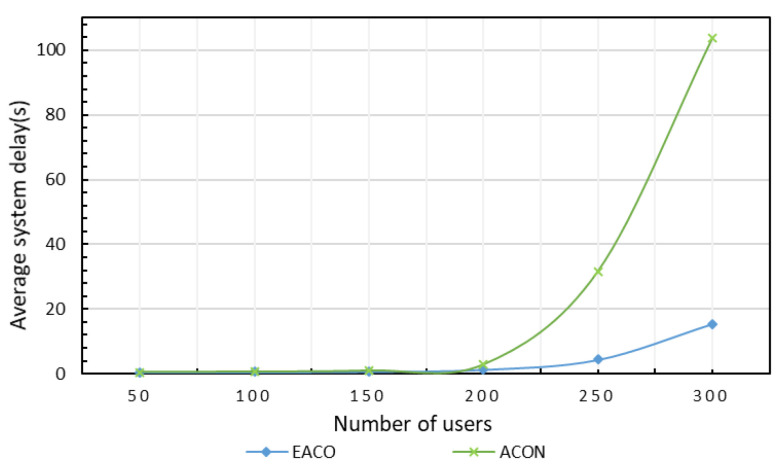
Comparison of system delay by different algorithms with/without mobile servers.

**Figure 6 sensors-23-06041-f006:**
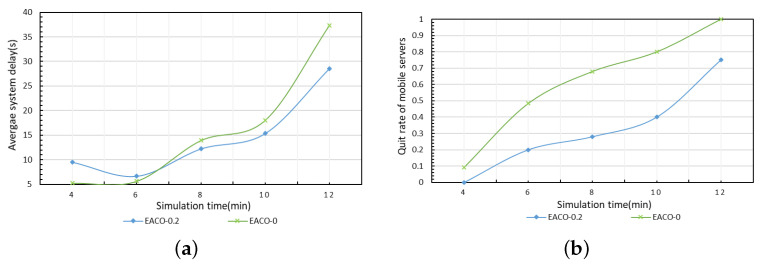
Comparison of performance on energy protection. (**a**) Average system delay. (**b**) Quit rate of mobile servers.

**Figure 7 sensors-23-06041-f007:**
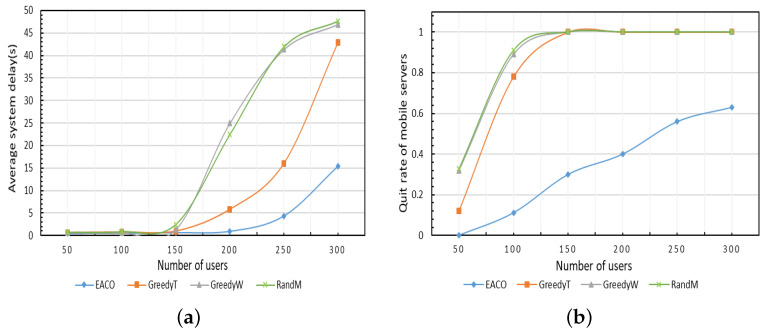
Impact of number of users. (**a**) Average system delay. (**b**) Quit rate of mobile servers.

**Figure 8 sensors-23-06041-f008:**
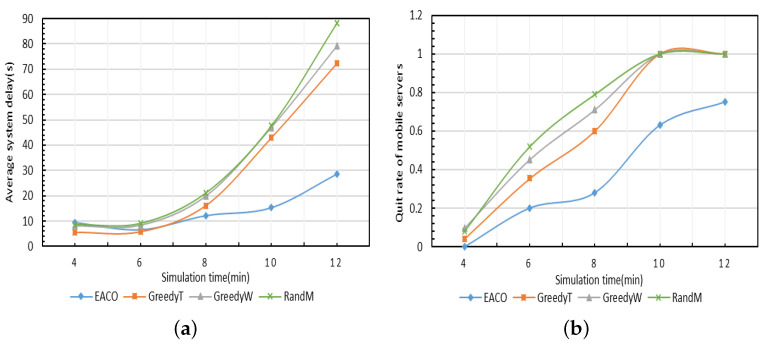
Comparison of performance of different algorithms. (**a**) Average system delay. (**b**) Quit rate of mobile servers.

**Table 1 sensors-23-06041-t001:** Main notations used.

Notations	Definitions
$a_{k,s}$	Binary variable indicating whether task $r_{k}$ is assigned to server s∈S for processing
$C_{k},D_{k},t_{k}$	Computation workload, data size and arriving time of task $r_{k}$
$T\;D_{k,s}$	The total delay of task $r_{k}$ from its offloading to being processed at server s∈S
K	Set of users
N	Set of fixed edge servers
$M,M_{n}^{t}$	“Set of all mobile servers” and “set of mobile servers associated with base station *n* in time cycle *t*”, respectively
$h_{k,n}^{t},h_{m,n}^{t}$	Binary variables indicating whether user *k* and mobile server *m* are associated with base station *n* in time cycle *t*, respectively
S	Set of all edge servers, including both fixed servers and mobile servers
$T_{k,s}^{com}$	The delay that task $r_{k}$ experienced at server s(s∈S), which includes the queuing delay and processing delay
$T_{k,s}^{trans}$	Transmission delay experienced by task $r_{k}$ on the way from user *k* to server s(s∈S)
$R^{t}$	Set of tasks arrived in time cycle *t*
$E_{m}^{max}$	Initial energy of mobile server *m*
$E_{m,k}^{cm}$	Energy consumed for mobile server *m* to process task $r_{k}$
$E_{m}^{t}$	Residual energy of mobile server *m* at the beginning of time cycle *t*
$E_{m}^{sel\;f}$	The amount of energy required to be reserved at mobile server *m* for its own use
$f_{s}$	Computing capacity of server *s*, s∈S
$P_{k,s}^{b}\left(g\right)$	Transition probability for ant *b* to choose vertex <k,s> in the *g*th round
$\tau_{k,s}\left(g\right)$	Pheromone value of vertex <k,s> in the *g*th round
α	Pheromone-value-related factor
β	Heuristic factor
$A^{Num},G$	Number of ants and number of iterations, respectively
$x_{s,wl}^{\delta }\left(r_{k}\right)$, $x_{s,lt}^{\delta }\left(r_{k}\right)$	Binary variables indicating whether the system is starting to transmit task $r_{k}\in R^{t}$ at time δ∈t via wireless and wired links of server *s*, respectively
$y_{k,k^{\prime }}^{wl}\left(s\right)$, $y_{k,k^{\prime }}^{lt}\left(s\right)$	Binary variables indicating the transmission sequence of task $r_{k}$ and task $r_{k^{\prime }}$

**Table 2 sensors-23-06041-t002:** Summarized performance.

EACO	ACO	EACO-0	Optimal Parameters
1	675%	103%	$A^{Num}=50$, ξ=0.2
**EACO**	**GreedyT**	**GreedyW**	**RandM**
1	253%	277%	309%

## Data Availability

Not applicable.

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
