# Peer review of "Energy Criticality Avoidance-Based Delay Minimization Ant Colony Algorithm for Task Assignment in Mobile-Server-Assisted Mobile Edge Computing"

_sensors, 2023, doi:10.3390/s23136041_

Round 1

Reviewer 1 Report

This paper presents task assessment and energy manamgent through ant colony optimization for the edge. The paper is filled with several equations. The evaluations measure the performance of the proposed approach, although it does not adequately benchmark against a base.

No comments on the language. The writing can be improved for clarity.

Author Response

Q1.1: This paper presents task assessment and energy manamgent through ant colony optimization for the edge. The paper is filled with several equations. The evaluations measure the performance of the proposed approach, although it does not adequately benchmark against a base.

A: Thank you for the comments! Actually, in our performance evaluation in the manuscript, we compared our proposed algorithm with three benchmark algorithms for comparison purpose, which are typically considered in practical scenarios. These three benchmark algorithms are as follows: a) Random assignment (RandM), which assigns each task randomly to a server; b) Minimum workload first assignment (GreedyW), which assigns each task to the server with the minimum workload among all choices; c) Minimum delay first assignment (GreedyW), which assigns each task to the server with the minimum delay among all choices.

  Following your advice, during the revision, we have added a set of new experiments (see Section 5.3.6 of the revised manuscript) to compare the performance of our proposed algorithm with other benchmark algorithms with varying simulation time.

Q1.2:Comments on the Quality of English LanguageNo comments on the language. The writing can be improved for clarity.

A: Thank you for the suggestion! We had carefully proofread the whole manuscript to eliminate grammar errors and typos and accordingly improve the quality of the presentation.

Reviewer 2 Report

- The abstract must be revised and improved.
- The Introduction / background section also needs to be improved with latest research work proposed in this research area since last 3 years.
- Authors should mention clearly about the novelty of the paper and contributions in the introductory section and in the abstract.
- Methodology section is well explained and easy for the reader to follow. 

Author Response

Q2.1: The abstract must be revised and improved.

A: Thank you for the valuable suggestion! During the revision, we have revised the Abstract to better highlight the major work in our manuscript and also eliminated the typos therein.

Q2.2:  The Introduction / background section also needs to be improved with latest research work proposed in this research area since last 3 years.

A: Thank you for the valuable suggestion! During the revision, we have added four latest research work in the Introduction section of the revised manuscript (see Refs. [1][2][14][15]). For more details, please refer to the revised manuscript.

Q2.3: Authors should mention clearly about the novelty of the paper and contributions in the introductory section and in the abstract.

A: Thank you for the suggestion! The novelty and contributions in this paper focus on proposing the mobile server assisted MEC architecture for improved task assignment performance, formulating the delay minimization problem under this architecture, proposing the dynamic energy criticality avoidance based delay minimization ant colony algorithm, deducing its computational complexity, and validating its effectiveness by comparing with different benchmark algorithms. All these points have been concisely pointed out and summarized in both the abstract and Introduction section. Thank you!

Q2.4 Methodology section is well explained and easy for the reader to follow.

A: Thank you for the good words.

Reviewer 3 Report

Dear Authors,

Very well done research, the topic is topical. However, I have a few remarks (two small and two quite important):

First remark (quite important): the paper has a skewed proportion structure, there is a huge research part and a very short Related work part. The title of part 2, indicates, one paper (work), when in the meantime it is a literature review, but it is still too short. I propose to expand pt. 2 : do a better literature review and change the title of pt. 2 into plural or simply call it: Literature review.

Second comment (minor) : in my opinion the main title: Joint Energy Management and Task Assignment for Mobile Servers Assisted Mobile Edge Computing, is too short and does not show the research warren, which is very good. In my opinion to the current title: Joint Energy Management and Task Assignment for Mobile Servers Assisted Mobile Edge Computing should be added: based on proposal methodology of dynamic Energy criticality avoidance based delayed minimization Ant Colony algorithm (EACO).

Third comment (minor):  the structure of para. 5.3 every time (5.3.1., 5.3.2 etc.) starts with a sentence about the drawing, in my opinion give a lot of text first and then a sentence talking about the drawing and the drawing.

Fourth comment (quite important): the conclusion is too short and does not show the whole essence of the algorithm and its applicability, as well as directions for further research.

Fifth comment (additional): I wonder whether to add a discussion before the conclusion. If other reviewers point out the lack of discussion, please do a discussion.

My final opinion: very nice methodology of the paper.

I wish you scientific success.

Reviewer

Author Response

Dear Authors,

Very well done research, the topic is topical. However, I have a few remarks (two small and two quite important):

A: Thank you for the good words!

Q3.1: First remark (quite important): the paper has a skewed proportion structure, there is a huge research part and a very short Related work part. The title of part 2, indicates, one paper (work), when in the meantime it is a literature review, but it is still too short. I propose to expand pt. 2 : do a better literature review and change the title of pt. 2 into plural or simply call it: Literature review.

A: Thank you for the valuable suggestion! In the revised manuscript, we have changed the title of Section 2 into “Literature Review” as suggested. In addition, we had further expanded this section to better reflect the state-of-the-art by incorporating more relevant references (i.e., [38]-[43]), which have different optimization objectives from the perspectives of energy use optimization and delay minimization. For more details, please refer to  Paragraph 3 of section 2 in the revised manuscript.

Q3.2:  Second comment (minor) : in my opinion the main title: Joint Energy Management and Task Assignment for Mobile Servers Assisted Mobile Edge Computing, is too short and does not show the research warren, which is very good. In my opinion to the current title: Joint Energy Management and Task Assignment for Mobile Servers Assisted Mobile Edge Computing should be added: based on proposal methodology of dynamic Energy criticality avoidance based delayed minimization Ant Colony algorithm (EACO).

A: Thank you for the valuable suggestion! During the revision, we have changed the paper title to “Energy Criticality Avoidance based Delay Minimization Ant Colony Algorithm for Task Assignment in Mobile Servers Assisted Mobile Edge Computing” to better reflect the major contribution in the manuscript.

Q3.3: Third comment (minor):  the structure of para. 5.3 every time (5.3.1., 5.3.2 etc.) starts with a sentence about the drawing, in my opinion give a lot of text first and then a sentence talking about the drawing and the drawing.

A: Thank you for the suggestion! During the revision, we have rephrased the descriptions of the paragraphs in Subsection 5.3. That is, for each paragraph, we first describe the objective of the corresponding test, then the simulation setting used, and finally report the results in the corresponding figure. We believe the rephrasing can largely ease the reading. Thank you!

Q3.4: Fourth comment (quite important): the conclusion is too short and does not show the whole essence of the algorithm and its applicability, as well as directions for further research.

A: Thank you for the valuable suggestion! In the revised manuscript, we have added some discussions on the essence and also properties of the proposed algorithm and finally pointed out some possible directions for further research in the conclusion section. Thank you again for the suggestion.

Q3.5: Fifth comment (additional): I wonder whether to add a discussion before the conclusion. If other reviewers point out the lack of discussion, please do a discussion.

A: Thank you for the suggestion! In the revised manuscript, we have added a summary on the  comparison of performance by different algorithms at the end of the simulation section. Hope the newly added contents can help the understanding of the effectiveness of the proposed algorithm in this manuscript.

Q3.6: My final opinion: very nice methodology of the paper.

           I wish you scientific success.

A: Thank you very much for the good words.

Reviewer 4 Report

The authors worked on task assignment and energy management in mobile servers-assisted edge computing networks. In particular, they considered server heterogeneity and energy constraints of mobile servers, aimed at minimizing the system delay while prolonging the serving time of mobile servers by energy use balancing. The work presented by the author is very interesting and the results support the conclusions. However, there are revisions that need to be addressed.

1. The scenario reported would benefit significantly from applying the device-to-device concept but the authors did not consider this in the projected design. Why?

2. To add more credibility to the simulation results, it would be more interesting to compare the results with actual measurements. What do you think?

3. I strongly recommend the addition of a comprehensive Table summarizing the key results and comparing them with the existing state-of-the-art.

4. There are occasional English construction issues that need to be revised. Please, check the English throughout the paper.

A minor English check is required.

Author Response

The authors worked on task assignment and energy management in mobile servers-assisted edge computing networks. In particular, they considered server heterogeneity and energy constraints of mobile servers, aimed at minimizing the system delay while prolonging the serving time of mobile servers by energy use balancing. The work presented by the author is very interesting and the results support the conclusions. However, there are revisions that need to be addressed.

A: Thank you for the good words!

Q4.1:  The scenario reported would benefit significantly from applying the device-to-device concept but the authors did not consider this in the projected design. Why?

A: Thank you for the comments and valuable suggestion! In this paper, we focused exclusively on the scenario where mobile servers work as the assistant processors for the fixed edge severs. Considering the authorization management and privacy protection, the transmissions of tasks to the mobile servers are made based on the relaying of base stations. For future directions, device-to-device task offloading can also be considered for further improved performance.

Q4.2:   To add more credibility to the simulation results, it would be more interesting to compare the results with actual measurements. What do you think?

A: Thank you for the comments! Indeed, it would be more convincing to compare the results with actual measurements. However, due to lack of sufficient infrastructure support and resources at this stage, we are unable to evaluate the performance of the proposed algorithm in a more practical system. For this reason, in this manuscript, we conduct extensive simulations to evaluate the performance of the proposed algorithm, a typically used validation method widely used in academia and industry. The simulation results validate the effectiveness of our proposed algorithm as compared with benchmark algorithms. We believe the results reported are sufficient to support the effectiveness of our proposed algorithm. In the future, when supportive experimental system is available, we would like to validate the performance of our algorithm in such environment to improve its practicality. Thank you again for the suggestion!

Q4.3: I strongly recommend the addition of a comprehensive Table summarizing the key results and comparing them with the existing state-of-the-art.

A: Thank you for the valuable suggestion! In the revised manuscript, we have added a new table (see Table 2 at the end of the “Performance Evaluation” section in the revised manuscript) to summarize the key simulation results for a comparison between our proposed algorithm and benchmark algorithms and architecture.  Besides, at the end of the “Literature Review” section in the revised manuscript, we have highlighted the major issues in existing work and how our work in this manuscript will address these issues. We hope this new table and also the motivation points summarized at the end of Section 2 in the revised manuscript can greatly ease the understanding of the effectiveness of the proposed algorithm in this manuscript and also how it differs from existing work. Thank you very much!

Q4.4: There are occasional English construction issues that need to be revised. Please, check the English throughout the paper.

A: Thank you for the suggestion! We had carefully proofread the whole manuscript to eliminate grammar errors and typos and accordingly improve the quality of the presentation.

Round 2

Reviewer 2 Report

accept